# Biologically Inspired Mechanisms for Adversarial Robustness

**Manish Reddy Vuyyuru**
Institute for Applied Computational Science
Harvard University
Cambridge, MA 02139
mvuyyuru@g.harvard.edu

**Andrzej Banburski**[*]
Center for Brains, Minds and Machines
Massachusetts Institute of Technology
Cambridge, MA 02139
kappa666@mit.edu

**Nishka Pant**[*]
Center for Brains, Minds and Machines
Massachusetts Institute of Technology
Cambridge, MA 02139
npant@mit.edu

**Tomaso Poggio**
Center for Brains, Minds and Machines
Massachusetts Institute of Technology
Cambridge, MA 02139
tp@ai.mit.edu

## Abstract

A convolutional neural network strongly robust to adversarial perturbations at reasonable computational and performance cost has not yet been demonstrated. The primate visual ventral stream seems to be robust to small perturbations in visual stimuli but the underlying mechanisms that give rise to this robust perception are not understood. In this work, we investigate the role of two biologically plausible mechanisms in adversarial robustness. We demonstrate that the non-uniform sampling performed by the primate retina and the presence of multiple receptive fields with a range of receptive field sizes at each eccentricity improve the robustness of neural networks to small adversarial perturbations. We verify that these two mechanisms do not suffer from gradient obfuscation and study their contribution to adversarial robustness through ablation studies.

## 1 Introduction

While modern convolutional neural networks (CNNs) have demonstrated remarkable performance in visual recognition, they still lag severely behind human vision on several tasks. Notably, despite intense effort in recent years, these models remain vulnerable to adversarial perturbations [1], even on standard datasets (e.g. ImageNet). In contrast, human vision appears to be particularly robust to small perturbations in visual stimuli.

Simultaneously, an increasing body of computational and experimental work demonstrates the suitability of these artificial neural networks (ANNs) as models of the primate brain's ventral visual stream [2, 3, 4, 5]. This suggests an opportunity to investigate if aspects of biological vision, that are not yet incorporated into current computer vision models, improve adversarial robustness. If the approach is successful, it will improve the robustness of engineered models and provide strong support for ANNs being good models of primate vision. If the proposed approach does not find any way to modify the existing models to reach human-level robustness, it could demonstrate that the broad family of feedforward deep networks should be rejected as models of primate vision.

**Engineering Significance** Robust real-world adversarial examples that fool models across a wide range of views, angles and lighting have been demonstrated [6]. CNNs are increasingly being

---

[*]A.B. and N.P. contributed equally to each other in this work.

considered for use in safety critical applications (e.g. autonomous vehicles), thus the dramatic sensitivity of state-of-the-art models to tiny perturbations in otherwise benign inputs has clear security implications. Proposed methods for improving adversarial robustness have generally approached the problem from either a prevention or detection perspective. Some of the proposed mechanisms include defensive distillation [7], feature squeezing [8], defensive randomization [9], adversarial training [10], and several have drawn inspiration from biological vision [11, 12]. Faithfully evaluating adversarial robustness has proven to be non-trivial [13, 14, 15], with many proposed methods eventually being shown to be ineffective at making models robust [16, 13, 17, 18]. Adversarial training remains the most promising at increasing the robustness of models. However adversarial training comes with several downsides, ranging from significantly increased computational cost, to preferential robustness against adversarial attacks that the model was trained on [17], to decreased standard accuracy [19] and ineffectiveness at improving robustness on examples in the low density regions of the training data distribution [20]. Mechanisms that help bring the models closer to human-level robustness and alleviate this tension between performance and security would have broad implications for the application of these networks.

The learning of feature representations that are particularly well suited to visual recognition tasks is a key characteristic of CNNs that has led to their widespread use. However, the features learned by current models that perform well in standard datasets can vary significantly from features that are meaningful to humans [21]. Recent studies have demonstrated that the quality of these learnt representations are improved in adversarially robust models. In particular, many aspects of feature visualization and manipulation were better aligned with our notion of visual perception in robust models [22, 23]. This suggests that adversarial robustness holds intrinsic value, beyond security implications and potentially even at the cost of standard accuracy, as a prior for helping models learn representations that are more human meaningful and interpretable.

**Biological Significance** Recent studies have argued that ANNs are suitable models of biological vision due to the similarity between internal representations in CNNs and in the primate brain [2, 4, 3]. In this paper, we explore two biological mechanisms that are not captured in current deep learning models of vision.

The uneven distribution of cones in the primate retina results in non-uniform spatial sampling of visual stimuli. The density of sampling is highest at a fixation point on an image and decreases with distance from the fixation point. In contrast, standard CNNs accept images sampled in a uniform square grid. Previous studies have demonstrated that incorporating the non-uniform sampling performed by the primate retina into standard networks improves predictivity of neural sites in the primate V4 cortical area, allows for better neural population control via controller images [3] and helps in generating adversarial examples that impact the accuracy of time-limited humans [24]. The first mechanism that we investigate incorporates information across multiple retinal fixation points.

The receptive field size along the primate visual stream increases with eccentricity [25, 26, 27]. From a sampling perspective, this translates to several scale-space image fragments centered on a fixation point in an image. Previous studies have argued for and demonstrated the computational role of pooling over the scale-space fragments in invariant visual recognition [28]. This second "multiple scales" mechanism allows for translation, scale and clutter invariance to be incorporated into neural networks [29, 30, 31], with much lower sample complexity than standard data augmentation methods. We investigate incorporating information across multiple "cortical fixations".

**Contributions** We demonstrate that two mechanisms inherent to primate vision (the "proposed mechanisms") consistently improve the adversarial robustness of neural networks to small adversarial perturbations across a range of PGD variants, hyperparameters and adversarial criteria (by about 0% to 30% for $\epsilon \leq 0.02$). One of the mechanisms (retinal fixations) improves robustness at almost no cost to standard performance (+1.62% on ImageNet). Through ablation studies, we also identified the key features of each mechanism that contribute to robustness. Our results suggest that biologically inspired mechanisms are promising candidates for improving robustness of standard neural networks. It also probes more broadly and provides support for the hypothesis that ANNs are good models of biological vision.

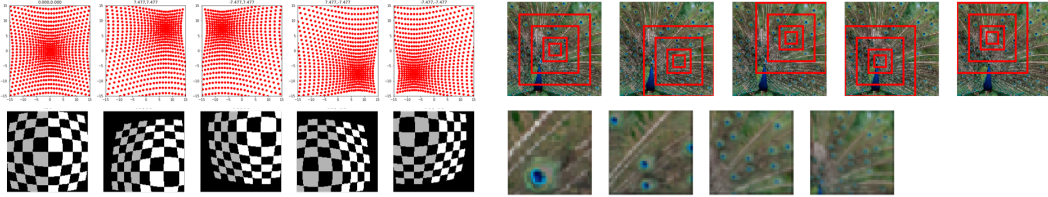

Figure 1: Left Top: Distributions of sampling points for 5 different retinal fixations. Red dots represent pixels that would be sampled from the original standard image to form the retina sampled image. Left Bottom: Effect of retinal sampling on an image of a flat checkerboard. Images presented were re-sampled at 5 different fixation points. Right Top: Shown in red, the centering of scale-space fragments on 5 different cortical fixations on the image. Right Bottom: The resulting 4 scale-space fragments from 1 fixation point. For a single fixation point, the 4 scale-space fragments result from crops of varying sizes but are all Gaussian downsampled to the size of the smallest.

## 2 Methods

**Datasets** The experiments were spread across 4 datasets. CIFAR10 is a small, standard dataset [32] and was used to benchmark results against other published results on adversarial robustness. The proposed mechanisms excel with images of much higher resolution. The majority of the experiments in this work were thus carried out on ImageNet10 and some experiments were repeated on ImageNet100 and ImageNet to study the scalability of the proposed mechanisms.

The ImageNet dataset [33] offers high resolution images split into 1000 classes that span a range of breadth (e.g. airliner, strawberry, etc.) and depth (e.g. mud turtle, leatherback turtle, etc.). 10 classes were hand picked to construct the ImageNet10 dataset. The classes were chosen to be visually distinct and of natural objects (e.g. pandas, snakes, etc.). To construct the ImageNet100 dataset, 100 classes were randomly chosen. Images in the ImageNet dataset are of varying dimensions. To standardize, all images in the ImageNet10, ImageNet100 and ImageNet datasets were formed only from the central 320x320 regions of the original images. For the full ImageNet dataset, in the interest of keeping the compute time for experiments reasonable, models were trained on the full training set but robustness evaluations were carried out on a test set downsampled to 5 images per class (totalling 5000 images for the 1000 classes). See Section 1 in the supplementary material for the preprocessing steps and the list of ImageNet classes comprising the ImageNet10 and ImageNet100 datasets.

**Biologically Inspired Mechanisms** The first mechanism is the non-uniform spatial sampling of visual stimuli by the photoreceptors. In the primate retina, the density of cones is maximum at the center of the fovea and decreases with eccentricity. The maximum density corresponds to a sampling distance of about 27 seconds of arc between adjacent cones, which corresponds almost exactly to the Shannon sampling limit imposed by the diffraction limited optics with a cutoff around 60 cycles/degree. The implementation for retinal sampling was adapted from [3] where it was tuned to mimic the exponentially decreasing density of cones with eccentricity. In [3], $g$ is defined as a function that maps points from the re-sampled image ($r'$) to the original image ($r$). As part of the transformation, the pixel coordinates in the image grid ($x, y$) were mapped to polar ($r, \theta$). To work with arbitrary fixation points, we re-centered this transform at a fixation point instead of at the Cartesian origin. We present here a visualization of the distribution of the sampling points (Figure 1: Top Left) and example retinal sampled images at 5 different fixation points (Figure 1: Bottom Left). See Section 2.1 in the supplementary material for additional details.

The size of receptive fields also varies with eccentricity, presumably to avoid aliasing [25, 26, 27]. The second mechanism is that V1 neurons show a range of scales at each eccentricity [28]. The main computational reason for this non-uniform sampling and the existence of a set of spatial scales is to enable processing of images with translation invariance – small but growing with larger receptive fields – and a large range of scale invariance. We assume here the estimate by [34] with 5 "frequency channels" having in the fovea receptive field with a diameter of $2s = 1'20"$, 3.1', 6.2', 11.7', 21', covering a range of roughly 1 to 20. Following from previous studies [28, 29, 30], we implemented the estimates for the set of scale-space fragments in V1 (i.e. the frequency channels estimated by [34]) by taking multiple crops that are progressively larger and Gaussian downsampling them to the dimensions of the smallest crop. We employed 4 scales of dimensions 40x40, 80x80, 160x160 and 240x240 that were all downsampled to 40x40 for ImageNet. For CIFAR10, we only employed 2 scales (15x15 and 30x30 that were both downsampled to 15x15) because of the significantly lower

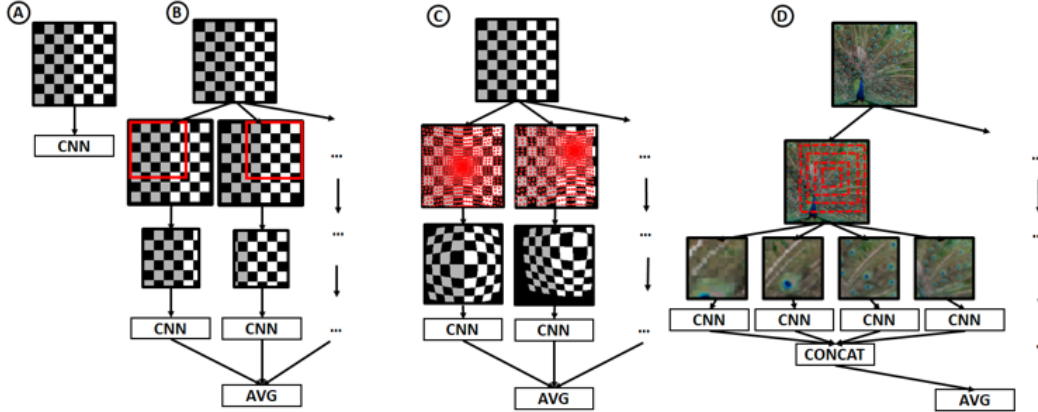

Figure 2: Model A (baseline): standard CNN. Model B (baseline): "coarse fixations". Model C (effect): "retinal fixations". Model D (effect): "cortical fixations". Red marks on images in second row indicate approximate action of mechanisms centered on a fixation point. 2 fixation points visualized for Models B and C, 1 fixation point visualized for model D.

resolution of images in the dataset (32x32 in CIFAR10 vs 320x320 for ImageNet). We present here a visualization of the centering of the scale-space fragments at various fixation points in an image (Figure 1: Top Right) and the resulting scales from one of the fixation points (Figure 1: Bottom Right). See Section 2.2 in the supplementary material for additional details.

When making a prediction for a given fixation point, models incorporating cortical sampling have to incorporate information across the different scales. Each scale is fed to a separate CNN which does not end with a dense layer. We concatenate the final latent vector from each branch, and this concatenated representation is then passed through a single dense layer for classification. During training time, we used an auxiliary loss to ensure that each branch was also predictive of the label on its own. See Sections 5.11 - 5.13 in the supplementary material for results from some preliminary experiments with alternate concatenation mechanisms (e.g. max, average pooling across scales, etc.).

**Models** Models for the CIFAR10 and ImageNet datasets were based off the standard CIFAR ResNet-20 and ImageNet ResNet-18 architectures [35].

We used a standard model and a "coarse fixations" model as baselines (Model A and Model B in Figure 2). The coarse fixations model approximates crudely the effect of fixations by applying a standard network to different image regions. This is effectively the same as standard 10-crop testing in [35] without image flipping.

The "retinal fixations" model applies a standard network to an image that is non-uniformly sampled as previously described (Model C in Figure 2). In the "cortical fixations" model, the standard ResNet architecture is split over multiple branches that process the scales at each eccentricity independently. The number of filters for each branch were chosen such that the total compute cost of a forward pass of all the branches costs about the same as a standard ResNet. The striding in early layers of the ResNet architecture was adjusted to account for the small dimensions of each scale-space fragment. We used an auxiliary loss during training to encourage information across all the scales to contribute to the model prediction (Model D in Figure 2).

During evaluation time, information was incorporated across fixations by averaging the model logits from 5 fixation points for each mechanism (top left, top right, bottom left, bottom right, middle).

**Training** The models for coarse, retinal and cortical fixations depend on a fixation point on the image to center the mechanism on. During training, a fixation point is randomly sampled from the valid set of fixations for the mechanism. This is in contrast to behavior at evaluation time, when information across fixations is incorporated by averaging model logits over 5 pre-determined fixation points for each mechanism. See Section 3 in the supplementary material for the valid and chosen fixations for each mechanism. Image augmentation was standardized across all models and datasets (random crops and random left/right flips). No color augmentation was used.

Models for CIFAR10 and ImageNet10 were both trained with an ADAM optimizer with $\beta_1 = 0.9$, $\beta_2 = 0.999$ and an initial learning rate of 0.001. CIFAR10 Models were trained for 200 epochs with a batch size of 180 and a fixed learning schedule (decay from initial by 0.1, 0.01, 0.001, 0.0005 at epoch 80, 120, 160, 180). Models for ImageNet10 were trained for 400 epochs with a batch size of 64 and a fixed learning schedule (decay from initial by 0.1, 0.01, 0.001, 0.0005 at epoch 160, 240, 320, 360). Models for ImageNet100 and Imagenet used an SGD optimizer with weight decay of 0.0001, momentum 0.9, initial learning rate of 0.1, and a batch size 256. Models for ImageNet100 were trained for 130 epochs with a fixed learning schedule (decay from initial by 0.1, 0.01, 0.001, 0.0005 at epoch 30, 70, 90, 120). Models for ImageNet were trained for 90 epochs with a fixed learning schedule (decay from initial by 0.1, 0.01, 0.001 at epoch 30, 60, 80).

**Adversarial Robustness** We followed previously proposed guidelines [13, 14, 15] when evaluating robustness. The adversarial attacks used were as implemented in the Python package Foolbox [36].

Projected Gradient Descent (PGD) [37] was motivated by previous works as a universal first order adversary that provides a suitable security guarantee against first-order attacks [10, 38]. In this work, the adversarial robustness of the proposed mechanisms was investigated primarily with PGD. PGD ($L_\infty$ variant) prescribes the generation of adversarial examples with the following iterative scheme:

$$x_{adv,i} = \text{CLIP}_{x,\epsilon}(x_{i-1} + \lambda \text{SIGN}(\nabla_x L(...))), \qquad x_{adv,0} = x_{original}$$

where $i$ is the count of iterations, CLIP is an operation that clips $x$ back to the permissible set, $\lambda$ is the step size and $\nabla_x L(...)$ is the gradient of the relevant loss function for the attack. The gradient was always fully propagated through the proposed mechanisms.

When setting a maximum perturbation size $\epsilon$ with PGD, we need to choose a distance metric. Various $L_p$ norms of the distance from the adversarial image to the original image are usually employed, $|x_{adv} - x_{original}|_p$, where typically $p = 2$ or $p = \infty$ [16]. The usual distance metrics are not necessarily well aligned with human perceived similarity. There is ongoing work in that area that would be interesting to consider in the future [39]. We mostly used $L_\infty$ PGD and $L_2$ PGD in our experiments, but also checked robustness to $L_1$ PGD and the fast gradient sign method (FGSM) [36].

We evaluated the robustness of the models by studying how the accuracy varied as $\epsilon$, the maximum perturbation size, was increased. We calculate the accuracy as 1 - ((num. naturally misclassified + num. adversarial examples) / num. images in test set). Whether an image is naturally misclassified was always determined by whether the true class was the most likely class predicted by the model, regardless of the adversarial criteria. Adversarial attacks were then only run against images that were not naturally misclassified. In most of the experiments, we set the step size $\lambda$ to $\epsilon/3$ and ran 5 iterations. This allows PGD to reach the edge of the permissible set and explore the boundary while keeping compute time for the experiments reasonable. We also conducted some experiments with 20 to 1000 iterations, step sizes $\lambda$ of $\epsilon/20$ to $\epsilon/3$, tried setting $\lambda$ dynamically with an ADAM optimizer, etc. See Section 4 and 5.1 in the supplementary material for additional details.

## 3  Results

We compared the test classification error to study standard performance. We refer to the performance of the models on the unperturbed natural test sets as the standard performance. The coarse fixations model generally outperformed the standard ResNet models, which is in line with previous studies [35]. The coarse fixations model underperformed on CIFAR10 likely due to the small size of images. The

Table 1: Standard TOP-1 performance of baselines and proposed models on various datasets.

| MODEL | CIFAR10 | IMAGENET10 | IMAGENET100 | IMAGENET |
|---|---|---|---|---|
| STANDARD RESNET | 88.13% | 89.4% | 76.80% | 59.46% |
| COARSE FIXATIONS | 87.70% | **91.2%** | **78.42%** | 61.28% |
| RETINAL FIXATIONS | **88.88%** | 90.2% | 78.22% | **62.90%** |
| CORTICAL FIXATIONS | 85.16% | 88.6% | 73.62% | 56.32% |

retinal fixations model performed about as well as the best baseline model (worse by only 0.2% on ImageNet100 and better by 1.62% on ImageNet) but the cortical fixations model decreased standard performance (underperforms by 4.96% on ImageNet). The relative performance of the models was fairly consistent across datasets. The performance penalties for the cortical fixations model were

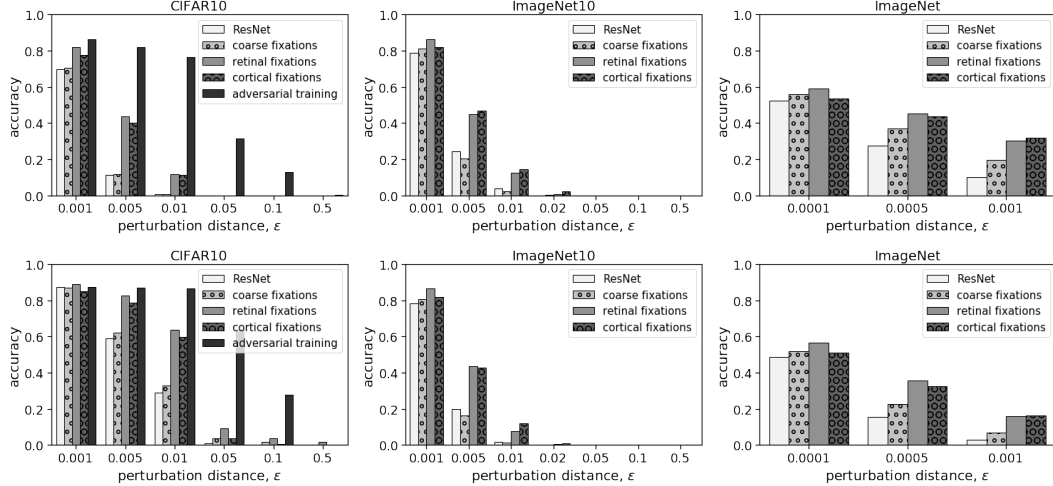

Figure 3: Robustness of the models on various datasets with increasing PGD perturbation budget. Top Row: robustness to 5-step $L_\infty$ PGD standard adversarial examples. Bottom Left: robustness to strongly misclassified adversarial examples, Bottom Middle: robustness to 20-step PGD, Bottom Right: robustness to $L_2$ PGD.

likely still less than for adversarial training. For example, $L_\infty$ PGD adversarial training on CIFAR10 decreases standard performance by about 7.9% [10].

## 3.1 Adversarial Robustness

The adversarial robustness of the proposed mechanisms was evaluated by studying the change in accuracy of the models with increasing perturbation budget. The experiments span a range of PGD variants, hyperparameters and adversarial criteria (Figures 3, 4).

The biologically inspired mechanisms consistently improved adversarial robustness to small perturbations across almost all experiments on all datasets (see Figure 4). The extent of improvement varied depending on the attack variants, hyperparameters and the dataset used. The retinal fixations and cortical fixations models improved robustness to about the same extent, with the best model varying across experiments. At larger perturbations, the mechanisms did not significantly impact robustness. See Section 5.9, 5.10 in the supplementary material for a preliminary inspection of the quality and visibility of the adversarial perturbations.

For small perturbations on ImageNet10, we observed a greater improvement in accuracy for the proposed models from the best baseline when using PGD as opposed to FGSM. This suggests that the mechanisms are effective at improving robustness, with the retinal and cortical fixations models performing proportionately better against stronger attacks that more thoroughly probe the robustness. The improvement in accuracy was relatively unchanged when increasing the number of iterations of PGD from 5 to 20 with step size $\lambda$ of 0.1 or 0.025. In general, the proposed models showed greater improvement in accuracy under $L_\infty$ PGD versus $L_2$ PGD, with the least improvement with $L_1$ PGD. Further studies could be conducted on variations in improvement with attack variants and hyperparameters to better understand the contribution of the mechanisms to adversarial robustness. See sections 5.2 - 5.8, 5.11 - 5.14 in the supplementary material for additional details.

**Confident Misclassifications** On ImageNet10, the robustness to standard misclassifications (true class not the most likely predicted class) was compared to the robustness to confident misclassifications (true class not in top 3 for untargeted attacks and adversarial class predicted with probability >80% for targeted attacks). Generally, the proposed mechanisms were more robust to confident mistakes than standard mistakes for untargeted attacks. The robustness to confident and standard mistakes for targeted attacks was about the same.

In a targeted setting, the loss used for PGD pushes towards a confident misclassification (objective is to increase probability of adversarial class) whereas in an untargeted setting, the loss does not explicitly push towards confident misclassifications (objective is to minimize true class, this can be accomplished either by increase probability of only 1 or any number of other classes). This suggests

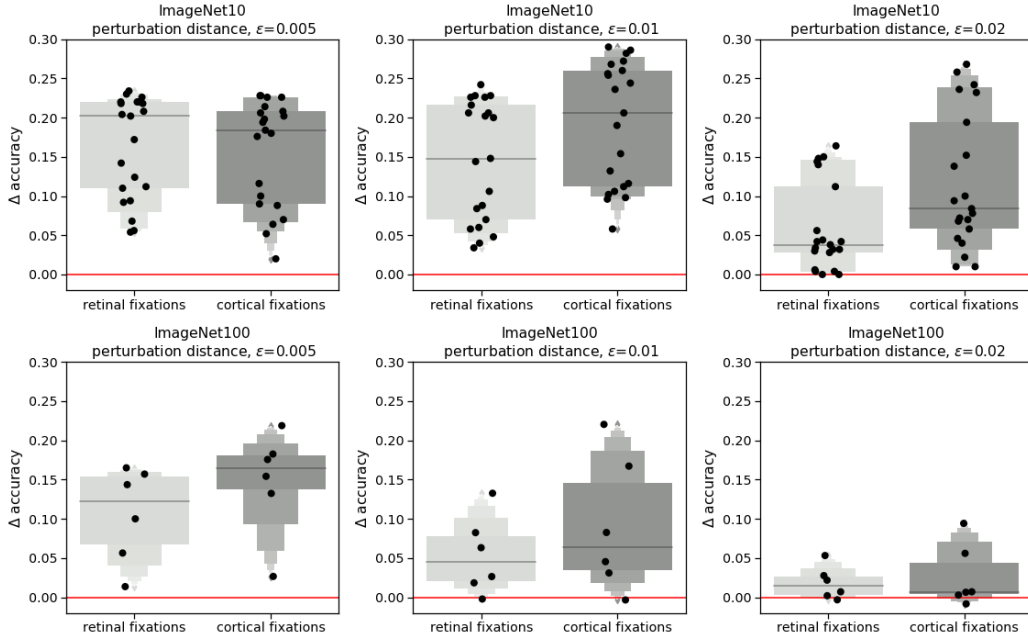

Figure 4: Improvement in accuracy of proposed models from the best baseline at small $\epsilon$. Each dot represents a different attack configuration. Top row: ImageNet10. Bottom row: ImageNet100.

that the proposed models improve robustness to confident misclassifications as confident mistakes were less likely to occur unless they were explicitly optimized for.

**Adversarial Training** On CIFAR10, the robustness of the models was benchmarked against an adversarially trained model available online from a previous study [10]. The model uses a variant of the standard ResNet architecture, w32-10 ResNet. We benchmarked the robustness of the models with $L_\infty$ PGD, which is also the attack that was used for adversarial training (Figure 3).

The robustness of the retinal fixations and cortical fixations models was consistently better than the baselines but worse than adversarial training. The accuracy of the models that were not adversarially trained dips to approximately 0% for perturbations of size $\epsilon = 0.05$ and higher. The proposed models made far fewer confident misclassifications than the baselines, with the robustness of the proposed models to confident misclassification at $\epsilon = 0.005$ approaching adversarial training.

## 3.2 Ablation Experiments

We performed experiments with a series of ablated models to study the contribution of various aspects of the proposed mechanisms to robustness. The experiments were conducted on ImageNet10 and span a range of PGD variants, hyperparameters and adversarial criteria (Figure 5). All ablated models employed a mechanism centered on fixations. As before, we incorporated information across fixations at evaluation time by averaging the model logits from 5 pre-determined fixation points in the image.

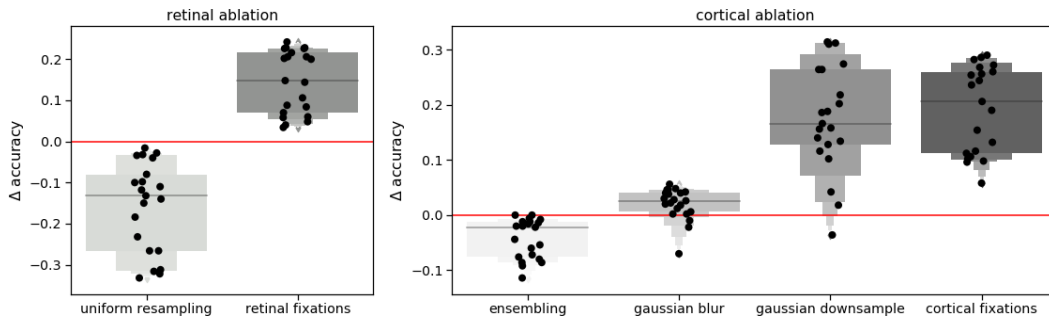

Figure 5: Improvement in accuracy of ablated models over the best baseline model under $L_\infty$ PGD with $\epsilon = 0.01$. Each dot represents a different attack configuration/criteria.

**Retinal Ablation** Non-uniform retinal sampling effectively sub-samples and up-samples the image. The pixels that get sub-sampled and up-sampled are chosen based on their distance to a point of fixation on the image. There are several factors that could potentially account for the improved robustness of the retinal fixations model, such as the sub-sampling or the sub-sampling combined with up-sampling or the non-uniformity of the sampling.

The retinal fixations model consistently outperformed the coarse fixations model in robustness tests. This indicates that simply sub-sampling, or applying a network on different image regions, does not improve robustness. We studied an ablated model that performs uniform sub-sampling (exactly as the coarse fixations model) followed by up-sampling. This ablated model ("uniform resampling" in Figure 5) consistently underperformed compared to the best baseline model. This indicates that uniformly sub-sampling and up-sampling an image does not improve (actually worsens) robustness.

This leaves the non-uniformity of the sampling as the main contributing factor to improving robustness. We speculate that the non-uniform sampling appropriately conditions the model to learn larger scale features during the training process and that these features are more robust to adversarial perturbations. This could be investigated further by studying the extent of robust versus non-robust features learnt by the model as in [21].

**Cortical Ablation** The cortical fixations model breaks an image into multiple scale-space fragments centered on a given fixation point. These fragments are constructed with a uniform square crop of increasing size followed by a Gaussian downsample of all fragments to the dimensions of the smallest fragment. The fragments are then fed through separate branches of a network and the linear classifier at the end of the network uses information from all branches. Potential factors contributing to the improved robustness include Gaussian blurs on sub-samples of the image, Gaussian downsampling on sub-samples of the image, classifying with a fixed computational budget distributed over separate branches of a network that combine their output, or a combination of the factors.

Classifying with a branched network ("ensembling" ablated model in Figure 5) worsened or at best did not change the robustness of the models. The "Gaussian blur" ablated model performs the sub-sampling exactly as the coarse fixations model, but applies a Gaussian blur on the images. This ablated model was more robust than the baselines but only to a small extent. It has been noticed previously that denoising operations (e.g. Gaussian blur) contribute to adversarial robustness [40, 41].

The "Gaussian downsample" ablated model is effectively the branch of the cortical fixations model that corresponds to the largest receptive field size. It performs sub-sampling similar to the coarse fixations model and applies a Gaussian downsample. This ablated model was more robust than the baselines but less robust than the cortical fixations model. This suggests that increasing reliance on large scale features in an image for classification partially contributed to the improved robustness of the cortical fixations models.

This leaves the ensembling across different scale-space fragments as the remaining contributing factor to the robustness of the cortical fixations model. It is possible that the fragments are sufficiently de-correlated to make ensembling effective, unlike in the case of the "ensembling" ablated model. We tried several versions of the cortical fixations model that incorporated information across the different scale-space fragments in various ways (see Section 5.12 in supplementary material). Some methods (e.g. pooling across branches) outperformed the current cortical fixations model in several experiments. This suggests that issues remain with properly incorporating features across the scale-space fragments in the cortical fixations model.

### 3.3   Gradient Obfuscation And Other Control Experiments

A common pitfall when evaluating adversarial robustness is gradient obfuscation [13]. By masking the gradient, models can appear to be robust under white-box attacks (e.g. PGD) which use the gradient to construct an adversarial example. We verified that the proposed mechanisms did not suffer from gradient obfuscation through previously suggested experiments [14]. Additionally, the measured robustness of models to adversarial attacks can be sensitive to the choice of attack hyperparameters such as the attack distance metric [17]. We compared the robustness of the models using a broad set of attack hyperparameters and also verified that our attacks had sufficiently converged [14]. See Sections 5.2 - 5.8, 5.14 and 6 in the supplementary material for additional details.

**0% Accuracy On Unbounded Attacks** With an unbounded adversarial attack, the accuracy of all models should always drop to 0. This is for the simple reason that an unbounded adversarial

perturbation can transform any image to any other image. Therefore, if the gradient information used in PGD is useful in constructing an adversarial example, the accuracy of any model should always be brought to 0. We observed this repeatedly when studying the accuracy of the models with increasing PGD perturbation budget. For example, the accuracy of all models drops to 0% for $\epsilon = 0.5$ under 5-step $L_\infty$ PGD on ImageNet10 (Figure 3).

**Iterative Attacks Outperform Single-Step Attacks** For models that do not suffer from obfuscated gradients, taking multiple steps repeatedly in the direction of gradient ascent should be more useful or at least as useful as taking a single step when generating adversarial examples [14]. We compared the robustness of the models against FGSM, which is exactly equivalent to a 1-step $L_\infty$ PGD, and against 5-step $L_\infty$ PGD for a range of $\epsilon$ from 0.001 to 0.5. The accuracy of the retinal fixations and cortical fixations models under PGD was always strictly lower or the same as under FGSM. This suggests that the models do not suffer from gradient obfuscation since PGD was more successful at finding adversarial examples. See Section 6.2 in the supplementary material for additional details.

**White-Box Attacks Outperform Black-Box Attacks** The gradient propagated through the models should be more useful for constructing adversarial examples than a crude approximation of the gradient. We compared PGD, which uses the exact gradient to construct adversarial examples (white-box attack), to three other attack algorithms that instead use a crude approximation of the gradient (black-box attacks). We constructed adversarial examples using a standard ResNet model as a substitute for the retinal fixations model and cortical fixations model (transfer attack). The examples generated with transfer attack heavily underperformed compared to PGD. In further experiments, PGD also outperformed boundary-attack [42] and backward pass differentiable approximation [13]. See Sections 6.3 in the supplementary material for additional details.

**Attack Hyperparameters** The measured robustness of models can be sensitive to the attack configuration used for evaluation [14]. For example, see [17]'s critique of [10] under $L_0$ or $L_2$ PGD. We studied the improvement in robustness with the proposed mechanisms using a broad set of hyperparameters. This set included various attack distance metrics ($L_1$, $L_2$, $L_\infty$), starting attacks with a random offset from the natural images, dynamic attack step sizes (ADAM), etc. Models were evaluated against each attack configuration using a range of $\epsilon$ from 0.001 to 0.5. Across almost all experiments, the proposed mechanisms were more robust to small perturbations than the best baseline model. See Sections 5.2 - 5.8 in the supplementary material for additional details.

**Attack Convergence** To keep the compute time reasonable, we used attacks with 5 or 20 steps in most of our experiments with step sizes of $\epsilon/3$ or $\epsilon/12$. To show that the trends in robustness seen in our experiments were representative of the general trend, we ran $L_\infty$ PGD for up to 1000 iterations using a step size of $\epsilon/20$ for a range of $\epsilon$ from 0.001 to 0.02. We demonstrated that despite greatly increasing the number of steps and decreasing the step size, the drop in accuracy for all models was minimal. This suggests that the attacks in our experiments with a smaller number of iterations and larger step sizes were already sufficiently converged and strongly representative of the general trend. See Section 5.6 in the supplementary material for additional details.

## 4    Conclusion

In this work, we showed that two key features of primate vision – foveation due to non-uniform distribution of cones in the retina and multiscale filtering because of receptive fields of different sizes in V1 at each eccentricity – consistently improve the robustness of neural networks to small adversarial perturbations. These mechanisms have negligible computational effect and one of them (non-uniform sampling) improves robustness at almost no cost in recognition performance. However, these mechanisms do not improve robustness to large perturbations. Preliminary inspection of the adversarial examples leads us to speculate that the inability of human perception to notice that the adversarial images are different from the normal ones breaks roughly around or above the transition in our experiments from "small" to "large" perturbation. In that case, our results suggest that the two mechanisms we identified may partially explain the robustness of primate vision to "small" perturbations, while an additional, separate mechanism – akin to an anomaly detector – may detect the presence of "large" perturbations.

In short, our results support the hypothesis that ANNs are suitable as core models of object recognition in primates. They also imply that implementing more biologically inspired mechanisms is promising for increasing adversarial robustness without forgoing standard performance.

## Acknowledgements

We thank Aleksander Madry and Fernanda De La Torre for illuminating discussions.

## Broader Impact

In terms of ethical aspects and future societal consequences, achieving adversarially robust models is of critical importance to deployment of autonomous technologies we can trust. Among some of the positive outcomes we could count autonomous vehicles immune to misleading by malicious agents, better trust in medical imaging applications and any other discipline to which AI can be applied and which requires safety guarantees. This naturally also comes with negative impacts, from reducing jobs available to humans, to potentially making surveillance technologies much more difficult to avoid, allowing authoritarian regimes a much tighter control over their citizens, as well as enabling progress in autonomous weapon systems. Additionally, every defense against adversarial attacks that has been so far proposed has eventually been found to be vulnerable. If such vulnerabilities are found in increasingly more accurate models of primate vision, this could suggest the possibility of the existence of dynamically changing adversarial attacks that would fool humans, leading to potentially new camouflage technologies.

## Funding Disclosure

Part of the funding is from Center for Brains, Minds and Machines (CBMM), funded by NSF STC award CCF-1231216, and part by a grant from Lockheed Martin.

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
