[Supplementary Material]

# Supplementary Material for Biologically Inspired Mechanisms for Adversarial Robustness

**Manish Reddy Vuyyuru**
Institute for Applied Computational Science
Harvard University
Cambridge, MA 02139
`mvuyyuru@g.harvard.edu`

**Andrzej Banburski**
Center for Brains, Minds and Machines
Massachusetts Institute of Technology
Cambridge, MA 02139
`kappa666@mit.edu`

**Nishka Pant**
Center for Brains, Minds and Machines
Massachusetts Institute of Technology
Cambridge, MA 02139
`npant@mit.edu`

**Tomaso Poggio**
Center for Brains, Minds and Machines
Massachusetts Institute of Technology
Cambridge, MA 02139
`tp@ai.mit.edu`

## 1 Datasets

### 1.1 Preprocessing

All images $x$ were centered on the 0 to 1 scale by $x/255$.

Images in ImageNet10 and ImageNet100 were formed from the central 320x320 regions of images from ImageNet. If the image dimensions for either height or width (or both) were less than 320, we upsampled the image with a bilinear filter with a constant aspect ratio so that both dimensions were at least 320x320.

We conducted a set of preliminary experiments on ImageNet10 with training images (not test images) trimmed to bounding boxes. Results from these preliminary experiments were not reported in the paper but we report the results here in the supplementary materials. The standard bounding boxes were used as provided with the ImageNet dataset. If images had 0 bounding boxes, they were discarded for this dataset. If images had 1 bounding box, they were trimmed to the bounding box before the standard central 320x320 crop described above was performed. If images had more than 1 bounding box, we only kept the first bounding box.

### 1.2 ImageNet10

Classes comprising the dataset are listed below:

1. Snake: n01742172 boa constrictor
2. Dog: n02099712, Labrador retriever
3. Cat: n02123045, tabby
4. Frog: n01644373, tree frog
5. Turtle: n01665541, leatherback turtle
6. Bird: n01855672 goose
7. Bear: n02510455 giant panda
8. Fish: n01484850 great white shark
9. Crab: n01981276 king crab
10. Insect: n02206856 bee

## 1.3 ImageNet100

Classes comprising the dataset are listed below:

['n01644900', 'n02096051', 'n04366367', 'n03544143', 'n02105412', 'n01914609', 'n02105162', 'n02132136', 'n03026506', 'n03063599', 'n02815834', 'n07802026', 'n01968897', 'n03788365', 'n04443257', 'n12998815', 'n02454379', 'n03991062', 'n04332243', 'n04254680', 'n02097298', 'n07590611', 'n03680355', 'n02165105', 'n01491361', 'n04120489', 'n03742115', 'n07880968', 'n02808304', 'n03888257', 'n03095699', 'n01494475', 'n03673027', 'n02488702', 'n01871265', 'n02104365', 'n02281787', 'n04118538', 'n01828970', 'n02837789', 'n03127747', 'n04005630', 'n02115913', 'n01514859', 'n03452741', 'n02107908', 'n01847000', 'n04200800', 'n04153751', 'n04389033', 'n02487347', 'n02769748', 'n01843383', 'n02219486', 'n02009912', 'n03676483', 'n02797295', 'n04417672', 'n04591157', 'n04229816', 'n02058221', 'n03814906', 'n02097130', 'n02939185', 'n03710637', 'n02116738', 'n04418357', 'n03775071', 'n04328186', 'n02090721', 'n02667093', 'n03929855', 'n02089078', 'n02389026', 'n03388183', 'n07613480', 'n02749479', 'n02174001', 'n07932039', 'n02112018', 'n02398521', 'n04069434', 'n03838899', 'n02233338', 'n03207743', 'n02791270', 'n02114855', 'n04204238', 'n02342885', 'n02110063', 'n01518878', 'n02099712', 'n01704323', 'n02168699', 'n04238763', 'n03494278', 'n03980874', 'n02097209', 'n01616318', 'n03131574']

## 2 Biologically Inspired Mechanisms

### 2.1 Retinal Sampling

See Section 2 in the main paper and consult Bashivan et al. (2019) for full details on the sampling procedure and chosen parameters. Code from Bashivan et al. (2019) was open-sourced at https://github.com/dicarlolab/retinawarp.

### 2.2 Cortical Sampling

Biological measurements (Gattass et al. (1981, 1988)) have demonstrated that in primates, the receptive field size varies with eccentricity. With the slope of the relationship between receptive field size and eccentricity also becoming steeper further down the ventral stream (Figure 6). See Section 2 in the main paper for full details on the implementation.

Figure 6: Biological measurements in V1, V2 and V4 support a model in which the maximum scale depends on eccentricity. Adapted from Freeman & Simoncelli 2011 (original monkey data from Gattass et al. (1981, 1988)).

# 3 Models

As described in the paper, we employed two baseline models ('ResNet' and 'coarse fixations') and two effect models ('retinal fixations' and 'cortical fixations'). At evaluation time, the 'coarse fixations', 'retinal fixations' and 'cortical fixations' models each center their mechanisms on 5 pre-defined fixation points (top left, top right, bottom left, bottom right, center) and combine information across fixations by averaging models logits across the fixation points.

The 'ResNet' baseline model directly feeds the full image through a standard ResNet architecture (32x32 for CIFAR10 or 320x320 for ImageNet). The 'coarse fixations' model applies a standard ResNet architecture to 5 different regions of the image (5 224x224 regions from the 320x320 image for ImageNet, centered on (0,0),(48,48),(48,-48),(-48,48),(-48,-48) or 5x24x24 from 32x32 for CIFAR10, centered on (0,0),(4,4),(4,-4), (-4,4), (-4,-4)). This acts as a very coarse approximation of foveation. At training time, the model randomly trains on a single region from all valid regions, not just the 5 to be used at evaluation time (i.e. centered on (rand(-48 to 48),rand(-48 to 48)) for ImageNet and centered on (rand(-4 to 4),rand(-4 to 4)) for CIFAR10).

The 'retinal fixations' model applies the retinal sampling at 5 different fixation points (re-samples the image, keeping dimensions the same). At evaluation time, the mechanism is centered on the points (0,0),(80,80),(80,-80),(-80,80),(-80,-80) for ImageNet and on the points (0,0),(8,8),(8,-8),(-8,8),(-8,-8) for CIFAR. At training time, the model randomly trains on a single fixation from all possible fixations (centered on (rand(-80 to 80),rand(-80 to 80) for ImageNet and centered on (rand(-8 to 8),rand(-8 to 8)) for CIFAR10).

The 'cortical fixations' model applies the cortical sampling at 5 different fixation points (re-samples the ImageNet images from 320x320 to 5x40x40 and CIFAR10 images from 32x32 to 2x15x15). At evaluation time, the mechanism is centered on the points (0,0),(40,40),(40,-40),(-40,40),(-40,-40) for ImageNet and (0,0),(1,1),(1,-1),(-1,1),(-1,-1) for CIFAR10. At training time, the model randomly trains on a single fixation from all possible fixations (centered on (rand(-40 to 40),rand(-40 to 40)) for ImageNet and centered on (rand(-1 to 1),rand(-1 to 1)) for CIFAR10).

# 4 Adversarial Robustness

## 4.1 Determinism In TensorFlow

We performed all experiments, training, etc. on TensorFlow 2.0.0. On this version, TensorFlow implements a reduced form of GPU-deterministic op functionality. We fix this to ensure deterministic operations with a patch from NVIDIA (see https://github.com/NVIDIA/tensorflow-determinism). This patch was applied only when evaluating adversarial robustness. We verified that the results from 3 experiments (coarse fixations, retinal fixations, cortical fixations models for 5 iterations of $L_\infty$ PGD with a targeted loss (y_adv = y_true + 1) with $\epsilon$=0.01 and step size of $\epsilon$ / 3) were consistent with and without this patch. See "check determinism" notebook in the checks folder in source code for additional details.

## 4.2 Accuracy Calculation

To calculate the accuracy under an adversarial attack, we employed the following procedure. We first evaluated the model on the natural images in the test splits for the datasets. This gives us a number of natural images that are correctly classified and a number that are misclassified (always determined by the true-class not being the top-1 class predicted by model, num_misclassified). We then ran the adversarial attack on the correctly classified natural images, generating some number of adversarial examples that meet the desired adversarial criteria (num_adversarial). We then calculated the accuracy under attack as 1-((num_misclassified+num_adversarial)/(total_num_images)).

# 5 Results

## 5.1 CIFAR10 Benchmark

We benchmarked our evaluation pipeline against the pipeline used in Madry et al. 2018 (see Table 1). We perform these benchmarks on the CIFAR10 test set of 10000 images. Across our experiments, our

Table 1: Comparison of accuracy under PGD attacks between our pipeline and the pipeline used in Madry et al. 2018. Top Table: 5-step $L_\infty$ PGD, step size of $\epsilon$/3. Bottom Table: 20-step $L_\infty$ PGD step size, step size of 2/255.

| $\epsilon$ | BENCHMARK ACCURACY | OUR ACCURACY |
|---|---|---|
| 0.001 | 86.15% | 86.149% |
| 0.005 | 82.06% | 82.06% |
| 0.01 | 76.60% | 76.60% |
| 0.05 | 31.26% | 31.26% |
| 0.1 | 12.95% | 12.949% |
| 0.5 | 0.12% | 0.1199% |

| $\epsilon$ | BENCHMARK ACCURACY | OUR ACCURACY |
|---|---|---|
| 8/255 | 45.72% | 45.73% |

pipelines were exactly consistent (except in the one case of 20-step PGD where our results differed by 1 image on the full test set). The benchmark pipeline from Madry et al. 2018 was open-sourced at https://github.com/MadryLab/cifar10_challenge.

## 5.2 All Experiment Hyperparameters

We provide an almost complete list of hyperparameters used for the experiments presented in the paper (Table 2). Please refer to the full evaluation data provided with the source code for an exhaustive list of all relevant hyperparameters. For ImageNet10 for example, not presented in the table, we also ran 1 experiment with 5 randomly chosen fixation points and 1 experiment where the images were initialized with random noise in an $\epsilon_{init}$ ball. Both these experiments were run with $L_\infty$ PGD for 5 steps with STEP_SIZE = 0.1 and CRITERION_TAG = MISCLASSIFY_3.

For each set of hyperparameters, the value for maximum perturbation size $\epsilon$ was scanned across [0.001, 0.005, 0.01 , 0.05 , 0.1 , 0.5 ] for CIFAR10 and [0.001, 0.005, 0.01 , 0.02, 0.05 , 0.1 , 0.5 ] for ImageNet.

## 5.3 Robustness To FGSM, PGD, PGD ADAM

We compared the robustness for the FGSM, PGD ADAM, PGD adversarial attacks using the $L_\infty$ distance metric, 5 attack iterations (1 for FGSM), step size of $\epsilon$ / 3 (5$\epsilon$ / 3 for FGSM), adversarial criteria of true class not in top-3 predicted classes by model at $\epsilon$ = [0.005, 0.01, 0.02]. Retinal fixations and cortical fixations models showed biggest improvements in robustness from the best baseline for PGD, and the least with FGSM (see Figure 7).

## 5.4 Robustness To PGD 5 Steps, PGD 20 Steps

We compared the robustness for $L_\infty$ PGD at 5 and 20 iterations. We ran the attacks with a step size of $\epsilon$ /3, with the adversarial criteria of true class not top-1 predicted class by model at $\epsilon$ = [0.005, 0.01, 0.02]. Improvement in robustness of retinal fixations and cortical fixations models from the best baseline was approximately the same for 5 and 20 iterations (see Figure 8).

## 5.5 Robustness To PGD Step Size $\epsilon$ /3, $\epsilon$ /12

We compared the robustness for $L_\infty$ PGD at step sizes of $\epsilon$ /3 (STEP_SIZE = 0.1) and $\epsilon$ /12 (STEP_SIZE = 0.025). We ran the attacks for 20 iterations, with the adversarial criteria of true class not top-1 predicted class by model at $\epsilon$ = [0.005, 0.01, 0.02]. Improvement in robustness of retinal and cortical fixations models from the best baseline was approximately the same for the 2 step sizes (see Figure 9).

Table 2: Attack hyperparameters used for evaluating the robustness of the models in most of the experiments presented in the paper. Please see full evaluation data provided with the source code for an exhaustive list. ATTACK_ALGO refers to the algorithm used for generating the adversarial example, DISTANCE_METRIC refers to the form of $L_p$ norm used to measure distances and the $L_p$ variant of the attack algorithm, ITERATIONS refers to the number of steps of the iterative algorithms, STEP_SIZE refers to a constant used internally to calculate the actual step size (actual step size: ($\epsilon$ / 0.3) * STEP_SIZE), CRITERION_TAG refers to the loss and criteria used for the attack (TARGETED_X: targeted attack (adv_class = true_class + 1, classified as adv_class with probability at least X if X != 50, classified as adv_class as top1 predicted class if X = 50), MISCLASSIFY_X: untargeted attack (true class not in top-X classes predicted by model)). Top Table: For experiments on CIFAR10, Middle Table: IMAGENET10, Bottom Table: IMAGENET100.

| ATTACK_ALGO | DISTANCE_METRIC | ITERATIONS | STEP_SIZE | CRITERION_TAG |
|---|---|---|---|---|
| PGD | LINF | 5 | 0.100 | MISCLASSIFY_1 |
| PGD | LINF | 5 | 0.100 | MISCLASSIFY_3 |

| ATTACK_ALGO | DISTANCE_METRIC | ITERATIONS | STEP_SIZE | CRITERION_TAG |
|---|---|---|---|---|
| PGD | L2 | 5 | 0.100 | TARGETED_50 |
| PGD | LINF | 20 | 0.025 | TARGETED_50 |
| PGD | LINF | 20 | 0.100 | TARGETED_50 |
| PGD | LINF | 5 | 0.100 | TARGETED_50 |
| PGD | L2 | 5 | 0.100 | MISCLASSIFY_1 |
| PGD | LINF | 20 | 0.025 | MISCLASSIFY_1 |
| PGD | LINF | 20 | 0.100 | MISCLASSIFY_1 |
| PGD | LINF | 5 | 0.100 | MISCLASSIFY_1 |
| FGSM | LINF | 1 | -1.000 | MISCLASSIFY_3 |
| PGD | L1 | 5 | 0.100 | MISCLASSIFY_3 |
| PGD | L2 | 5 | 0.100 | TARGETED_80 |
| PGD | L2 | 5 | 0.100 | MISCLASSIFY_3 |
| PGD | LINF | 20 | 0.025 | TARGETED_80 |
| PGD | LINF | 20 | 0.025 | MISCLASSIFY_3 |
| PGD | LINF | 20 | 0.100 | TARGETED_80 |
| PGD | LINF | 20 | 0.100 | MISCLASSIFY_3 |
| PGD | LINF | 5 | 0.100 | TARGETED_80 |
| PGD | LINF | 5 | 0.100 | MISCLASSIFY_3 |
| PGD_ADAM | LINF | 5 | 0.100 | MISCLASSIFY_3 |

| ATTACK_ALGO | DISTANCE_METRIC | ITERATIONS | STEP_SIZE | CRITERION_TAG |
|---|---|---|---|---|
| PGD | L2 | 5 | 0.100 | MISCLASSIFY_1 |
| PGD | LINF | 5 | 0.100 | MISCLASSIFY_1 |
| PGD | L2 | 5 | 0.100 | MISCLASSIFY_3 |
| PGD | LINF | 5 | 0.100 | MISCLASSIFY_3 |
| PGD | L2 | 5 | 0.100 | MISCLASSIFY_10 |
| PGD | LINF | 5 | 0.100 | MISCLASSIFY_10 |

## 5.6 Robustness To PGD Up To 1000 Steps, Step Size $\epsilon/20$

To demonstrate that the trends in robustness observed with PGD at 5 or 20 iterations with step sizes of $\epsilon/3$ or $\epsilon/12$ were already representative of the general trend, we compared the accuracy for $L_\infty$ PGD at 1000 and 300 iterations against accuracy at 20 iterations. We ran the attacks with 1000 and 300 iterations with step size of $\epsilon/20$. We ran the attack with 20 iterations with step size of $\epsilon/12$. We ran these attacks on the ImageNet10 dataset. Despite the many additional steps and smaller step size, the observed drop in accuracy for all the models was minimal. The accuracy did not drop significantly for any of the models with increasing number of iterations up to 1000. The attacks with a smaller number of iterations and larger step size were already sufficiently converged and already strongly representative of the general trend (see Table 3).

Figure 7: Improvements in robustness for retinal fixations and cortical fixations models from the best baseline under various adversarial attacks at small perturbations on ImageNet10.

Figure 8: Improvements in robustness for retinal fixations and cortical fixations models from the best baseline at varying PGD attack iterations at small perturbations on ImageNet10.

## 5.7 Robustness Under $L_\infty$, $L_2$, $L_1$ Metrics

We compared the robustness for $L_\infty$, $L_2$ and $L_1$ PGD run for 5 iterations with a step size of $\epsilon$ /3 and the adversarial criteria of true class not in the top-3 predicted classes by model at $\epsilon$ = [0.005, 0.01, 0.02]. Improvement in robustness of retinal and cortical fixations models was greater for $L_\infty$ and least for $L_1$ (see Figure 10).

## 5.8 Robustness To Confident Mistakes

We compared the robustness for $L_\infty$ PGD run for 5 iterations with a step size of $\epsilon$ /3 and the adversarial criteria of untargeted misclassified out of top-1 or top-3 predicted classes by model (misclassify_1, misclassify_3) and targeted misclassification (y_adv = y_true + 1) with adversarial class in top-1 or predicted with at least 80% probability (targeted_50, targeted_80). Improvement in robustness of retinal and cortical fixations models was greater for misclassify_3 compared to misclassify_1. Improvement in robustness of retinal and cortical fixations models was approximately the same for targeted_50 and targeted_80 (see Figure 11).

Figure 9: Improvements in robustness for retinal fixations and cortical fixations models from the best baseline at varying PGD attack steps at small perturbations on ImageNet10. STEP_SIZE = 0.1, 0.025 corresponds to a PGD step size of $\epsilon/3$ and $\epsilon/12$.

Figure 10: Improvements in robustness for retinal and cortical fixations models from the best baseline using various distance metrics at small perturbations on ImageNet10.

Table 3: Comparison of accuracy under $L_\infty$ PGD with up to 1000 steps on ImageNet10. Due to time and computational constraints, we did not run $L_\infty$ PGD on the cortical fixations models for 1000 steps with $\epsilon = 0.001$. However, up to 500 steps, the accuracy stayed at 82%, with matched performance for bigger $\epsilon$. Standard: standard ResNet, Coarse: coarse fixations, Retinal: retinal fixations, Cortical: cortical fixations.

| $\epsilon$ | STANDARD | STEPS = 20, COARSE | STEP SIZE = $\epsilon/3$ RETINAL | CORTICAL |
|---|---|---|---|---|
| 0.02 | 0% | 0% | 0.4% | 1% |
| 0.01 | 1.6% | 1.2% | 7.6% | 11.8% |
| 0.005 | 20% | 16.4% | 43.4% | 42.8% |
| 0.001 | 78.4% | 80.8% | 86.4% | 82% |

| $\epsilon$ | STANDARD | STEPS = 300, COARSE | STEP SIZE = $\epsilon/20$ RETINAL | CORTICAL |
|---|---|---|---|---|
| 0.02 | 0% | 0% | 0.2% | 0.8% |
| 0.01 | 1.4% | 0.8% | 7% | 9.4% |
| 0.005 | 19.4% | 15.8% | 43.4% | 42.2% |
| 0.001 | 78.4% | 80.8% | 86.4% | 82% |

| $\epsilon$ | STANDARD | STEPS = 1000, COARSE | STEP SIZE = $\epsilon/20$ RETINAL | CORTICAL |
|---|---|---|---|---|
| 0.02 | 0% | 0% | 0.2% | 0.8% |
| 0.01 | 1.4% | 0.8% | 6.8% | 9.4% |
| 0.005 | 19.2% | 15.4% | 43.4% | 41.8% |
| 0.001 | 78.4% | 80.8% | 86.4% | - |

## 5.9 Quality of Adversarial Perturbations

We explored how semantically meaningful ("quality") large adversarial perturbations were for cortical fixations and retinal fixations models (see Figure 12). The large adversarial perturbations for the models were not anymore meaningful than for the coarse fixations model.

## 5.10 Visibility of Adversarial Perturbations

We explored the $L_\infty$ distance at which the adversarial perturbations become visible to the naked eye when the adversarial example and the original image are presented side by side. We present the visualized scan for an image of a shark here (see Figure 12). At preliminary inspection, the adversarial perturbations seem to become visible around $\epsilon \approx 0.2$ or 0.5.

## 5.11 Training With Only Bounding Boxes

We tried training the cortical fixations model on just the training images bounding boxes. We trimmed images in the training set of ImageNet10 (only training images, test images were left as is) to just their first bounding box and trained a cortical fixations model without the standard auxiliary loss during training. We compared this to a cortical fixations model trained on the usual training images without the standard auxiliary loss during training (see Figure 13).

Since only some of the images in the training set come with a bounding box annotation and since we only kept the first bounding box annotation in images, the model trained on the bounding boxes trained on far fewer samples. The model trained only on the bounding boxes did not show improved robustness compared to the model trained on the standard images.

## 5.12 Alternate Cortical Sampling Implementations

We carried out some experiments with alternate methods of incorporating information across the scales (that result from a single fixation point) for the cortical sampling (see Figure 14). We experimented with max and average pooling across the scales and employing a large dropout during training on the concatenated representation.

Figure 11: Improvements in robustness for retinal and cortical fixations models from the best baseline for various adversarial criteria at small perturbations on ImageNet10.

For the cortical fixations models as used in the rest of the paper, we simply concatenated the representation from each scale to form the complete representation for each fixation. For the pooling alternatives, we pool the representations across the scales. For the large dropout alternative, we randomly drop 75% of the elements in the final concatenated representation during training time. These alternate methods of incorporating information across the scales all performed better than the cortical sampling as used in the rest of the paper. This suggests that there are more robustness improvements to be gained from the cortical sampling models.

## 5.13 Combining Retinal And Cortical Sampling

We attempted to combine the two proposed mechanisms, by first performing the retinal sampling on the image, then the cortical sampling. We envisioned this to be similar to how when presented with a visual stimulus, the re-sampling by the photoreceptors would occur before any further processing in the visual stream. Naively combining the two mechanisms in this manner resulted in an improvement in robustness from the baselines but the combined mechanisms model was only as robust as or less robust than either of the proposed mechanisms (see Figure 15).

Figure 12: Original image and adversarial examples at various $L_\infty$ distances $\epsilon$. Adversarial examples were generated for the cortical fixations model (top row), retinal fixations model (middle row), coarse fixations model (bottom row) using 5-step PGD with step size $\epsilon$ /3.

Figure 13: Robustness of cortical fixations models trained on only the bounding boxes of training images ('w. bbox') and on the standard training images. Both models were trained without the standard auxiliary loss during training. Both models were evaluated on ImageNet10 to targeted (y_adv = y_true + 1) $L_\infty$ PGD attacks run for 5 iterations with step size $\epsilon$/3.

Figure 14: Robustness of alternate cortical fixations models as compared to the cortical fixations model as used in the rest of the paper on ImageNet10 to targeted (y_adv = y_true + 1) $L_\infty$ PGD attacks run for 5 iterations with step size $\epsilon/3$.

Figure 15: Robustness of the retinal fixations models as compared to the cortical fixations model and the combined (retinal and cortical) fixations model on ImageNet10 to targeted (y_adv = y_true + 1) $L_\infty$ PGD attacks run for 5 iterations with step size $\epsilon/3$.

Table 4: Accuracy of the models under $L_2$ CarliniWagner attack with the initial constant set to 0.01 on ImageNet10. Retinal and cortical fixations models outperformed the baselines (standard ResNet, coarse fixations).

| MECHANISM | ACCURACY |
|---|---|
| VANILLA RESNET | 0.692 |
| COARSE FIXATIONS | 0.71 |
| RETINAL FIXATIONS | 0.81 |
| CORTICAL FIXATIONS | 0.758 |

## 5.14 L2 CarliniWagner Attack

We also conducted some preliminary experiments comparing the robustness of the models to the L2 variant of the CarliniWagner attack (Carlini & Wagner (2018)) (see Table 4). We ran the attack for 100 iterations with the initial constant set to 0.01 for the ImageNet10 models. The preliminary results are consistent with our reported results with PGD (proposed mechanisms outperform the baselines).

## 6 Gradient Obfuscation

### 6.1 Expectation Over Random Transformations

A proper evaluation of models with random behavior at testing time requires taking an expectation over the gradient at each attack iteration (Athalye et al. (2018)). The coarse, retinal and cortical fixations models can be used as stochastic models. The transformation performed by the mechanisms at training time was random. However, at evaluation time, the transformations were pre-determined and fixed. Thus, there was no need for taking expectations over the transformation in almost all of our experiments. We conducted a single set of experiments with random transformations at evaluation time. For that experiment, we took an expectation of the gradient from 5 random samples of the transformation at each iteration of the attack.

### 6.2 Iterative Attacks Outperform Single-Step Attacks

To check that gradient obfuscation was not an issue with the models, we verified that PGD was always more successful or as successful as FGSM when generating adversarial examples (see Table 5). We verified this with $L_\infty$ PGD run for 5 iterations against the equivalent FGSM, with the adversarial criteria that the true class is not in the top-3 classes predicted by the model.

We saw that PGD consistently outperformed or at the very least performed as well as FGSM, this suggests that following the direction of gradient descent is useful for constructing adversarial examples and that gradient obfuscation is likely not an issue with the models.

### 6.3 White-Box Attacks Outperform Black-Box

If gradient obfuscation is not an issue with the models, white-box attacks (attacks that directly use the model gradients) should always outperform black-box attacks.

#### 6.3.1 Transfer Attack

We performed a transfer attack from the standard ResNet model to the retinal fixations and cortical fixations models (see Table 6). We compared the accuracy of the models under this transfer attack and accuracy under standard PGD. We perform $L_\infty$ PGD for 5 steps, with a step size of $\epsilon/3$, maximum adversarial perturbation $\epsilon = 0.005$ or $\epsilon = 0.5$. We perform targeted attacks with desired adversarial class ($y + 1 \mod n$, $n = 10$) and desired predicted probability of at least $0.8$.

For each perturbation limit, we first run PGD on the images in the ImageNet10 test split that were correctly predicted by the standard ResNet model. We then evaluate the retinal fixations and cortical fixations models on these adversarial examples generated for the standard ResNet model (tarnsfer attack). For the comparison to an equivalent white-box attack, we then run PGD on the retinal

Table 5: Comparison of accuracy under PGD and equivalent FGSM attack. Top Table: Retinal Fixations Model. Bottom Table: Cortical Fixations Model. PGD consistently performs as well as or outperforms FGSM.

| $\epsilon$ | ACCURACY (FGSM) | ACCURACY (PGD) |
|---|---|---|
| 0.500 | 0.054 | 0.000 |
| 0.100 | 0.240 | 0.000 |
| 0.050 | 0.378 | 0.014 |
| 0.020 | 0.540 | 0.236 |
| 0.010 | 0.698 | 0.600 |
| 0.005 | 0.850 | 0.840 |
| 0.001 | 0.900 | 0.900 |

| $\epsilon$ | ACCURACY (FGSM) | ACCURACY (PGD) |
|---|---|---|
| 0.500 | 0.108 | 0.010 |
| 0.100 | 0.326 | 0.036 |
| 0.050 | 0.440 | 0.098 |
| 0.020 | 0.636 | 0.344 |
| 0.010 | 0.742 | 0.638 |
| 0.005 | 0.834 | 0.818 |
| 0.001 | 0.886 | 0.886 |

Table 6: Accuracy of the retinal fixations and cortical fixations models was lower under a standard PGD attack performed directly on the models compared to a transfer attack from a standard ResNet model.

| MECHANISM | ATTACK METHOD | ACCURACY |
|---|---|---|
| RETINAL | PGD ($\epsilon = 0.005$) | 0.560694 |
| RETINAL | TRANSFER ($\epsilon = 0.005$) | 0.933333 |
| CORTICAL | PGD ($\epsilon = 0.005$) | 0.549133 |
| CORTICAL | TRANSFER ($\epsilon = 0.005$) | 0.887179 |
| RETINAL | PGD ($\epsilon = 0.5$) | 0.000000 |
| RETINAL | TRANSFER ($\epsilon = 0.5$) | 0.181208 |
| CORTICAL | PGD ($\epsilon = 0.5$) | 0.000000 |
| CORTICAL | TRANSFER ($\epsilon = 0.5$) | 0.261745 |

fixations and cortical fixations models for the images in ImageNet10 test split that were correctly predicted by both the standard ResNet model and the retinal or cortical fixations model.

The transfer attacks consistently underperformed at generating adversarial examples compared to a direct white-box PGD attack. This suggests that gradient obfuscation is not an issue with the models.

### 6.3.2 Boundary Attack, 1000 Steps

We compared the adversarial examples generated with boundary attack (black-box decision-based attack, Brendel et al. 2018) with PGD (white-box gradient-based attack) (see Table 7).

Both attacks were run in a targeted setting on models trained on ImageNet10. We ran it on a single image of a panda (ILSVRC2012_val_00042615.JPEG) to be misclassified as a shark. We initialized the boundary attack with an image of a shark (ILSVRC2012_val_00029481.JPEG). We only ran the comparison for a single image because boundary attack takes a while to run. We ran boundary attack for 1000 iterations and PGD for 5 iterations, comparing the size of adversarial perturbations.

Generated adversarial perturbations were always smaller (better) with 5 steps of PGD compared to 1000 steps of boundary attack. This again suggests that gradient obfuscation is not an issue with the models. The size of adversarial perturbations for the retinal and cortical fixations models generated with boundary attack was also larger than for the coarse fixations or standard ResNet model.

The probability assigned to the mispredicted class for examples generated with 5 steps PGD was also much greater than for 1000 steps of boundary attack ($\approx 100\%$ for PGD vs $\approx 50\%$ for boundary

Table 7: Size of adversarial perturbations generated with 5 steps of PGD was always less than with 1000 steps of boundary attack. Here, we refer to the standard ResNet model as 'vanilla', coarse fixations as 'coarse', retinal fixations as 'retinal', cortical fixations as 'cortical'.

| MECHANISM | ATTACK METHOD | DISTANCE |
|---|---|---|
| CORTICAL | PGD | 8.2E-06 |
| CORTICAL | BOUNDARY ATTACK | 2.5E-03 |
| RETINAL | PGD | 1.0E-05 |
| RETINAL | BOUNDARY ATTACK | 1.4E-03 |
| COARSE | BOUNDARY ATTACK | 1.1E-03 |
| VANILLA | BOUNDARY ATTACK | 6.45E-04 |

Figure 16: $L_\infty$ distance from image to adversarial example and assigned probability to mispredicted class against index in the ImageNet10 dataset for 10 images. Comparing 10000 steps of boundary attack and 20 steps of PGD. Top: using the retinal fixations model. Bottom: using the cortical fixations model.

attack). This is however an unfair comparison since PGD explicitly optimizes for the probability of the mispredicted class while boundary attack does not, but we mention it here for completeness.

### 6.3.3 Boundary Attack, 10000 Steps

Additionally, to demonstrate that the results with 1000 iterations of boundary attack with a single image were already representative of the general trend, we also compared 10000 iterations of boundary attack against 20 iterations of PGD on 10 images from ImageNet10. For all images, the size of adversarial perturbations (via $L_\infty$ distance) generated with boundary attack was larger than with PGD. The probability assigned to the mispredicted class for examples generated with PGD was also greater than for boundary attack.

### 6.3.4 Backward Pass Differentiable Approximation

We also attempted to generate adversarial examples using the backward pass differentiable approach (BDPA) (Athalye et al. (2018)) (see Table 8). If gradient obfuscation is not an issue with the models, this surrogate gradient approach should not outperform directly using the full model gradient with PGD.

Table 8: Ability to find an adversarial example with standard PGD was always superior to the BPDA PGD. Top, Bottom Tables: Ability to find adversarial examples using various hyperparameters with two different pictures as the original image.

| MECHANISM | ATTACK METHOD (STEPS, $\epsilon$) | ADV FOUND |
|---|---|---|
| CORTICAL | PGD (5,0.005) | Y |
| CORTICAL | PGD BPDA (60,0.005) | N |
| CORTICAL | PGD BPDA (5,0.5) | Y |
| RETINAL | PGD (5,0.005) | Y |
| RETINAL | PGD BPDA (120,0.005) | N |
| RETINAL | PGD BPDA (5,0.5) | Y |
| MECHANISM | ATTACK METHOD (STEPS, $\epsilon$) | ADV FOUND |
| CORTICAL | PGD (5,0.01) | Y |
| CORTICAL | PGD BPDA (300,0.01) | N |
| CORTICAL | PGD BPDA (5,0.5) | Y |
| RETINAL | PGD (10,0.005) | Y |
| RETINAL | PGD BPDA (300,0.005) | N |
| RETINAL | PGD BPDA (10,0.5) | Y |

For the retinal fixations model, we approximated the retina transform $g(x)$ as $g(x) \approx x$ for the gradient (backward pass). The cortical fixations model performs several transforms $h(i(j(x)))$ for each scale fragment. We approximated $i(x)$ as $i(x) \approx x$ for the gradient (backward pass). We did not approximate the gradients for the subsample and cropping operation further. We performed $L_\infty$ PGD on one image of a panda (top in Table 8) and one image of a frog (bottom in Table 8).

Standard PGD, taking the gradient through the full model, was always superior to PGD BPDA at finding an adversarial example. This suggests that gradient obfuscation is not an issue with the models.

We also verified that PGD BDPA was able to find adversarial examples when we allow very large perturbations.

## 7   Image Sources

The original pictures of the peacock (e.g. see right side of Figure 1 in main paper) and checkerboard (e.g. see bottom left side of Figure 1 in main paper) used to visualize the effect of the biologically inspired mechanisms were obtained from online sources. The image of a peacock was taken from Flickr (see `www.flickr.com/photos/kkoshy/32401990166`). The author, Koshy Koshy (see `https://www.flickr.com/photos/kkoshy/`), explicitly authorized the image for reuse and modification under the Attribution 2.0 Generic (CC BY 2.0) license (`https://creativecommons.org/licenses/by/2.0/`). The image of a checkerboard was taken from Wikimedia Commons (see `https://en.m.wikipedia.org/wiki/File:Checkerboard_reflection.svg`). The author, M. W. Toews (see `https://commons.wikimedia.org/wiki/User:Mwtoews`), explicitly authorized the image for reuse and modification under the Attribution-ShareAlike 4.0 International (CC BY-SA 4.0) license (`https://creativecommons.org/licenses/by-sa/4.0/deed.en`).