[Reviews · NeurIPS 2020]

Review 1

Summary and Contributions: Two neuroscience-inspired mechanisms are proposed to increase network robustness to adversarial attacks. “Retinal fixations” stands for a non-uniform sampling of the visual field around the fixation point (densest close to the center); “Cortical fixations” is a multi-scale processing by smaller branches of the network. Both mechanisms seem to work to some extent for small perturbations. According to the authors, this robustness breaks for perturbations visible to humans.

Strengths: Empirical evaluation of a typical ResNet architecture (for the multi-scale approach, divided into branches processing each scale) on three image datasets (Cifar10, ImageNet10, ImageNet100 and ImageNet) provides evidence that the proposed mechanisms increase network robustness. The topic is well matched with the NeurIPS community interests.

Weaknesses: The robustness is still much smaller than that achieved by adversarial training and it comes at a cost (although the Retinal Fixations out-performed the baseline on two datasets). All results are shown as a single number, undermining the confidence in the size of the effects shown, although the trend towards robustness was consistent across all tasks.

Correctness: Increasing the number of trained networks to provide the statistics for the observed effects would strongly improve the paper. I would also suggest to include results for perturbation distance = 0 in Fig. 3, to ease the evaluation of the cost of applying the proposed mechanisms. Please, correct the definition of accuracy (L163).

Clarity: Yes, but please provide more references to the Supplementary Material.

Relation to Prior Work: A quick google search for multi-scale processing yields a few citations within the topic of adversarial robustness. Given the plethora of mechanisms proposed in this field, I would encourage the authors to more thoroughly review the existing literature.

Reproducibility: Yes

Additional Feedback: I would encourage the authors to shortly describe the exact implementation of the proposed mechanisms, as the main contributions of the paper (or at least refer the reader to the Supplementary Material, where I eventually found some description). POST REVIEWS: Thank you for the comprehensive response. I stand by my point that it is a good submission, specifically interesting to the neuroscience part of the NeurIPS community.


Review 2

Summary and Contributions: 1. The authors study using two biological-inspired mechanisms, retinal fixations and cortical fixations, for improving adversarial robustness. Th2. e authors suggest that the non-uniform distribution of cones and multiscale filtering improve robustness to small adversarial perturbation at almost no extra cost.

Strengths: 1. Interesting biological-motivated mechanism to go beyond standard uniform-sampling. 2. Figure 5 shows informative ablation study for the improvement.

Weaknesses: 1. Given adversarial accuracy is an upper bound of true robustness, I am not sure whether the adversarial attack in the experiments is strong enough to truly evaluate the robustness. In Figure 3, it is only using 5-20 step PGD (no random restarts, not running for longer steps). It might not be sufficient. I am not sure whether I should trust the results and analysis from the weak attack. 2. Suppose the evaluation reflects the model robustness, the region for “small perturbations” is quite small. In CIFAR10, the proposed mechanisms only work on par with adversarial training at epsilon=0.001. I would think even epsilon=8/255 on CIFAR10 are not perceptible by humans. 3. In Table 1, it will be interesting to integrate the proposed fixations with SOTA models. The reported RESNET result is not the SOTA ones. 4. In the gradient obfuscation part, it will be good to clarify what does the authors “verified”. 5. Last paragraph in the conclusion generalizes the work a bit too much. 6. There is no adversarial training baseline in ImageNet10, ImageNet results.

Correctness: See the weakness section.

Clarity: The paper is clearly written.

Relation to Prior Work: It might be worth mentioning another related work [1]. Zoran et al., Towards Robust Image Classification Using Sequential Attention Models, 2020

Reproducibility: Yes

Additional Feedback: --- Update after author response: I have read the response. I think it is an interesting new perspective to the robustness filed, though I still find it insufficient to claim "robustness to adversarial attacks". Hence I increase the score slightly.


Review 3

Summary and Contributions: The paper tries to understand the mechanisms that potentially make human vision robust to test-time adversarial attacks. Two biologically plausible mechanisms are considered: 1) Retinal fixation: The first mechanism models the non-uniform sampling of the image performed by the retina due to uneven distribution of cones. This mechanism essentially involves subsampling and upsampling the pixels of the image. The density of sampling is highest at a fixation point on an image and decreases with distance from the fixation point. The final predicted output is the average of predicted output for subsampled images with different fixation points. 2) Cortical fixation: This mechanism models the presence of multiple receptive field sizes. It essentially involves setting a fixation point on the image, creating cropped images centered at this fixation spot with various sizes, downsampling the larger cropped images so that all the cropped images have the same size, and feeding these cropped images to a classifier. Final predicted output is the average of the prediction for different fixation points. The paper empirically shows that adding these mechanisms before feeding the input to neural networks help improve their adversarial robustness, as compared to standard training and some other baselines. The experiments are done on CIFAR-10 and various subsets of Imagenet dataset. However, it does not perform as well as defenses such as adversarial training where the model is explicitly trained to be robust. --------------------------------------------------------- Update after author response: I have read the author response. I feel this paper is an interesting attempt, but some of my concerns mentioned in the weaknesses still remain. Therefore, I am keeping my score the same.

Strengths: While the problem of coming up with defenses against adversarial examples has received significant attention, much less effort has gone into understanding what makes human vision seemingly so robust. In that sense, this paper is relatively unique and it does a decent job of trying to fill this gap. Moreover, it can potentially lead to more future work in this direction.

Weaknesses: 1) It is not very clear why the mechanisms proposed in the paper help with robustness. For example, an ablation study conducted in the paper shows that uniform subsampling of pixels hurt robustness while non-uniform subsampling seems to help. Given such small changes to the sampling procedure have such opposite effects on robustness, it makes it difficult to derive useful insights from the results of the paper. A more thorough investigation to understand the reasons behind the efficacy of the proposed mechanisms can result in a much more convincing paper. 2) The proposed mechanisms only improve the robustness marginally as compared to standard training. As discussed in the paper conclusion, this suggests that these mechanisms might not capture the first-order reasons for the apparent robustness of human vision. In that sense, the results in the paper might be of limited use for understanding the primary reasons for the robustness of human vision.

Correctness: Seems correct to me.

Clarity: Yes

Relation to Prior Work: Yes

Reproducibility: Yes

Additional Feedback:


Review 4

Summary and Contributions: in this paper, the authors propose two CNN architectures that are biologically inspired and show evidence that both architectures increase robustness against adversarial examples. Both architecture process the input image at different "fixation" locations and average the predictions. The first architecture (retinal fixations) samples in a retinal, i.e., nonuniform way around the fixation locations. The second architecture (cortical fixations) processes the input image at multiple resolutions around each fixation. The authors test the robustness of their architectures on CIFAR10, ImageNet10, ImageNet100 and ImageNet against the PGD attack.

Strengths: - I very much like the idea to take inspiration from the human visual system to increase the robustness of DNNs. This is a topic of great interest in terms of scientific and application impact to the NeurIPS community. - Although I see several weaknesses in the robustness evaluation (see below), I am happy to see that the authors invested a substantial amount of work into a range of control experiments and followed best-practices better than in many other papers. - the authors provide an extensive supplement with details on all experiments

Weaknesses: Robustness evaluations depend crucially choosing correct hyperparameters. Here I see several problems: - The authors mainly test robustness against the PGD attack with 5 or 20 steps. In general, this is way too little. To increase confidence in the results, I would need query success curves in the style of Brendel et al, NeurIPS 2019, where they show attack success rates over a logarithmic range of steps reaching up to 1000. Attack rates are often decreasing till the highest number of steps. Such query success curves can show whether the attack has already converged or not for a given number of steps - the authors don't provide justification for the chosen learning rates for PGD. The mostly used value of epsilon/3 seems very rough. I would expect them to test a substantially smaller learning rate (let's say epsilon/20) with substantially more iterations (>=250). Better even, test a substantial range of learning rates for each tested number of steps to find the optimal settings. - a lot of crucial control experiments, while conducted, are barely or not at all mentioned in the manuscript (l 269-291). - tests of decision based attacks are way too little (only one image, only 1000 steps) to show that there is no gradient obfuscation. Figure 4 in the boundary attack paper (Brendel et al, ICLR 2018) shows that even after 10000 steps the attack is still making progress.

Correctness: I didn't find any correctness issues beyond the weaknesses I described above.

Clarity: - the paper is overall well structured and written. Especially the model architecture is well explained - the additional experiments described in lines 269-291 contain way to little details. Most importantly, there is not even a reference to the supplement, which I found to contain all the missing details. - line 271: It appears that the paragraph "Gradient obfuscation" is parallel to the following paragraphs and it is not clear to the reader that the next paragraphs describe the experiments the researchers conducted as suggested in ref [12].

Relation to Prior Work: - the paper discusses only very little related work when it comes to drawing inspiration from biology for improving robustness. One example is https://arxiv.org/abs/1703.09202, but there is also a substantial body of work on improving robustness with attention mechanisms, which is closely related to the present work but neither mentioned nor compared to. Just two examples that are already published are Zoran et al, "Towards Robust Image Classification Using Sequential Attention Models" or Hendrycks et al "Using Self-Supervised Learning Can Improve ModelRobustness and Uncertainty". I'm aware of many more papers on arxiv where I'm not sure about the publication status.

Reproducibility: Yes

Additional Feedback: For me, the main issue are the weak evaluations of the robustness which decreases confidence in the actual robustness. For increasing confidence in the robustness, I would suggest the authors - run PGD with many more steps and smaller step size, including query-success-curves to judge convergence - run the boundary attack on a larger subset of images and with many more steps (and report how PGD performs on the same subset of images to make results comparable) If then the robustness results still hold, the authors make a much stronger argument for the robustness of their method and I would be very happy to improve my rating of the paper. --------- Update after the rebuttal phase I'm very happy to see that the authors have addressed all of my major concerns. The robustness evaluation still could be even more thorough, e.g., using random restarts. However, unlike most other defense methods, the proposed sampling methods in the paper to me seem to be quite well conditioned (no obvious gradient obfuscation or noise sources etc), therefore I don't expect substantial changes with restarts. I'm a bit surprised that the Boundary attack is not reaching the same performance as the PGD attack (with enough steps, it should be at least as good), but given the additional results, I'm sufficiently happy that gradient obfuscation is not a problem here. Even if the robustness might be not state-of-the-art, I think the biological inspiration of the presented method makes the results relevant nevertheless. Therefore I'm increasing my rating from "okay but not good enough, reject" to "marginally above acceptance threshold".

[Author Response · NeurIPS 2020]

We thank the reviewers for their insightful and positive feedback. We are very encouraged they enjoyed (R2, R4) the
idea to take inspiration from the human visual system to increase robustness of DNNs and found it to be unique (R3) and
worthy of investigating more. The reviewers appreciated the substantial amount of control experiments and following
best practices better than many other papers (R4), including showing scaling between datasets of varying complexities
(R1). We are pleased R2 found the ablation studies informative. We appreciate the constructive comments and questions
from the reviewers, some on the organization of the text between the main part and the extensive (R4) Supplementary
Material, and some suggesting further experimental tests to strengthen the results. We especially would like to thank
R4 for suggesting several very relevant papers to cite. As we were constrained by limited time and computational
resources, as well as the space to answer, we focus below on the most crucial additional experiments and questions, but
will incorporate all feedback in the final version.

**(R2, R4) Run PGD with many more steps and smaller step size to truly evaluate robustness.** The reviewers raised
concerns that our main tests of robustness were against the PGD attack with 5 or 20 steps with a large step size of
$\epsilon/3$ (we would like to point out that we have verified robustness to smaller step size of $\epsilon/12$ in the Supp. Mat.). Both
reviewers suggested to run attacks with smaller step size and follow (Brendel et al, NeurIPS 2019), where attacks were
performed up to a 1000 steps. We have followed this suggestion and performed PGD attacks on the ImageNet10 dataset
with step size of $\epsilon/20$ (as suggested by R4) and over a range of steps up to 1000, as shown in the Table below. The
drop in robustness is minimal with the additional steps. The trend of robustness seen with the small number of steps
was already strongly representative of the general trend, and we believe these additional results help drive the message
home that both retinal and cortical sampling are biologically-motivated methods that significantly improve robustness at
small perturbations without noticeable computational overhead.

Table 1: PGD with more steps on ImageNet10. Standard: standard ResNet; Coarse: coarse fixations on standard ResNet; Retinal: Retinal fixations model; Cortical: cortical fixations model (Due to time constraints, we were unable to finish running the 1000 step attack at $\epsilon = 0.001$, but up to 500 steps the accuracy stayed at 82%, with matched performance for bigger $\epsilon$).

| | STEPS = 20, STEP SIZE = $\epsilon/3$ | | | | STEPS = 300, STEP SIZE = $\epsilon/20$ | | | | STEPS = 1000, STEP SIZE = $\epsilon/20$ | | | |
| $\epsilon$ | STANDARD | COARSE | RETINAL | CORTICAL | STANDARD | COARSE | RETINAL | CORTICAL | STANDARD | COARSE | RETINAL | CORTICAL |
| --- | --- | --- | --- | --- | --- | --- | --- | --- | --- | --- | --- | --- |
| 0.02 | 0% | 0% | 0.4% | 1% | 0% | 0% | 0.2% | 0.8% | 0% | 0% | 0.2% | 0.8% |
| 0.01 | 1.6% | 1.2% | 7.6% | 11.8% | 1.4% | 0.8% | 7% | 9.4% | 1.4% | 0.8% | 6.8% | 9.4% |
| 0.005 | 20% | 16.4% | 43.4% | 42.8% | 19.4% | 15.8% | 43.4% | 42.2% | 19.2% | 15.4% | 43.4% | 41.8% |
| 0.001 | 78.4% | 80.8% | 86.4% | 82% | 78.4% | 80.8% | 86.4% | 82% | 78.4% | 80.8% | 86.4% | x% |

**(R4) Run boundary attack on a larger subset of images and with many more steps.** The reviewer suggested that
evaluating the boundary attack on a single image for 1000 iterations is insufficient to claim avoiding gradient obfuscation.
While we would claim that it served it's purpose together with many other tests reported in the main text and shown
in Supp Mat., we have nonetheless ran the attack on more images for up to 10000 steps and compared it to PGD on
the same subset. Since these attacks were extremely computationally expensive, we were only able to run them on 10
images for the 10k steps. So far, the trend agrees with the observation made in the paper (see Figure below). In all
cases, the size of the perturbation was smaller with PGD, while the probabilities assigned to the mispredicted class for
examples generated with 20 steps PGD were much greater than for 10k steps of boundary attack.

**(R3) Marginal improvement.** The reviewer critiques that our proposed mechanisms only improve the robustness
marginally as compared to standard training. While this is true for large perturbations (which both R3 and we note
suggests additional mechanisms), the effect at small perturbations is strong: at $\epsilon = 0.005$ the adversarial performance
more than doubles over the baselines both in CIFAR10 and ImageNet10.

**(R1) Comparison to adversarial training.** While the improvement from both mechanisms is lower than that of
adversarial training, at $\epsilon = 0.005$ on CIFAR10 they reach approximately the halfway point between standard models
and those adversarially trained, as seen in Figure 3 in the paper. However, unlike adversarial training, which is very
computationally expensive, our methods pose negligible computational overhead and actually lead to improvement of
standard performance on some of the datasets with the retinal fixations – this should be compared to adversarial training,
which provably leads to significantly worse standard performance. Additionally, we have shown in the Supp. Mat. that
our methods can scale to larger datasets (see the ImageNet100 experiments), while so far adversarial training has been
unsuccesful in being scaled up to larger number of classes (hence the suggestion of R2 to try adversarial training on
ImageNet is beyond the scope of this work).

[Meta-Review · NeurIPS 2020]

The paper was heavily discussed among all the reviewers, and in the end, the reviewers greed that the contributions of the paper are sufficiently significant (in terms of the scientific method and insights to the computational neuroscience community). The paper provides two neuroscience-inspired mechanisms are improve robustness of NNs to adversarial attacks. “Retinal fixations” stands for a non-uniform sampling of the visual field around the fixation point (densest close to the center); “Cortical fixations” is a multi-scale processing by smaller branches of the network. Both mechanisms seem to work to some extent for small perturbations. The reviewers also had a number of suggestions which will certainly improve the quality of the results (esp. to the ML audience). The suggestions are all mentioned in the "updated" reviews (e.g. more thorough evaluation).